# New Insights into the Mechanism of *Trichoderma virens*-Induced Developmental Effects on *Agrostis stolonifera* Disease Resistance against Dollar Spot Infection

**DOI:** 10.3390/jof8111186

**Published:** 2022-11-10

**Authors:** Lu Gan, Yuelan Yin, Qichen Niu, Xuebing Yan, Shuxia Yin

**Affiliations:** 1College of Animal Science and Technology, Yangzhou University, Yangzhou 225009, China; 2Institute of Turfgrass Science, Beijing Forestry University, Beijing 100107, China

**Keywords:** *Trichoderma virens*, metabolome, transcriptome, *Trichoderma*–plant interaction, dollar spot, defense responses

## Abstract

*Trichoderma* is internationally recognized as a biocontrol fungus for its broad-spectrum antimicrobial activity. Intriguingly, the crosstalk mechanism between the plant and *Trichoderma* is dynamic, depending on the *Trichoderma* strains and the plant species. In our previous study, the *Trichoderma virens* 192-45 strain showed better pathogen inhibition through the secretive non-volatile and volatile substrates. Therefore, we studied transcriptional and metabolic responses altered in creeping bentgrass (*Agrostis stolonifera* L.) with *T. virens* colonization prior to a challenge with *Clarireedia homoeocarpa*. This fungal pathogen causes dollar spot on various turfgrasses. When the pathogen is deficient, the importance of *T. virens* to the enhancement of plant growth can be seen in hormonal production and microbe signaling, such as indole-3-acrylic acid. Therefore, these substrates secreted by *T. virens* and induced genes related to plant growth can be the ‘pre-defense’ for ensuing pathogen attacks. During *C. homoeocarpa* infection, the *Trichoderma*–plant interaction activates defense responses through the SA- and/or JA-dependent pathway, induced by *T. virens* and its respective exudates, such as oleic, citric, and stearic acid. Thus, we will anticipate a combination of genetic engineering and exogenous application targeting these genes and metabolites, which could make creeping bentgrass more resistant to dollar spot and other pathogens.

## 1. Introduction

Dollar spot is one of the most prevalent turfgrass diseases in the world, caused by the pathogen *Clarireedia* spp. Various cool- and warm-season turfgrasses are at risk from extensive damage by dollar spot, including creeping bentgrass (*Agrostis stolonifera* L.), bermudagrass (*Cynodon dactylon* Pers.), Kentucky bluegrass (*Poa pratensis* L.), fescue (*Festuca* spp.), zoysiagrass (*Zoysia* spp.), etc. [1]. Creeping bentgrass (*Agrostis stolonifera* L., As) is one of the most important turfgrasses and has huge economic value in golf sports in the temperate regions of the world [2]. The quality of the golf game is always influenced by the height, texture, firmness, and other performance of the grasses [3]. Dollar spot disease always result in substantial economic losses and accounts for a significant portion of the costs of fungicides. For example, each spray targeting dollar spot costs USD 356.00/ha [4]. Damage from the disease has a major financial impact and affects the appreciation and playability of turfgrass, in terms of exposed land and weeds. Although integrated approaches, including resistant varieties, cultural practices, and fungicide applications, were effectively applied to control dollar spot under field conditions, frequent fungicides are still popular to manage the disease [5,6]. Repeated use of fungicidal compounds leads to decreased field efficacy of fungicides and the emergence of resistant populations [7,8]. In addition, there are few reports about the application of *Trichoderma* on turfgrass and pasture diseases.

The excessive use of chemical pesticides has caused several negative impacts on the environment and human health. Residues of chemical pesticides persist in soil for a time, destroying the soil ecosystem and increasing toxicity to plants and underground water pollution [9,10]. *Trichoderma* is a well-known eco-friendly and the most widely used biological fungicide and plant growth regulator in agriculture [11]. Its long-term persistence in the soil makes *Trichoderma* an effective disease control agent for vegetables, crops, and grasses [12]. The commercial strain of *Trichoderma harzianum* 1295-22 is capable of preventing lawn brown spots caused by *Rhizoctonia solani* and dollar spot caused by *Sclerotinia homoeocarpa* [13,14]. *Trichoderma ghanense* strain GCPL175 has an extraordinary restraining effect on *Lepiota* strain 1506, which causes mushroom rings on golf courses [15]. *Trichoderma* also improves plant growth, development, and biomass of turfgrass and pasture. Xie et al. found that *Trichoderma harzianum* had a noticeable enhancement on the vegetative growth of *Festuca rubra* and *Poa pratensis* [16]. These points suggest *Trichoderma* can be a viable and sustainable alternative to chemical fertilizers and pesticides for turfgrass management.

*Trichoderma*-based biocontrol is a varied mechanism mediated by mycoparasitism, production of antigenic metabolites and enzymes, competition for nutrients and rhizosphere occupation, induction of plant defense, promotion of plant growth, and resistance to stress conditions [17,18,19,20,21,22]. In some cases, more than one mechanism is involved in the interaction between plants and *Trichoderma*. *Trichoderma* species are known for their diverse bioactivity due to their abundance of secondary metabolites [23]. Hundreds of metabolites produced by *Trichoderma* have been isolated and characterized, including peptaibols, terpenes, steroids, amides, and others [24]. These compounds not only exhibited antifungal activity (such as harzianolide) but also possess the properties of a plant growth-enhancer and bioinducer [22]. The primary and secondary metabolites produced by *Trichoderma* could act as microbe-associated molecular patterns (MAMPs) to stimulate the defense system of the plants, such as trichothecenes, indole-3-acidic acid or their derivatives, and polyphenols [25,26,27]. Although *Trichoderma* has been widely studied, more metabolites will likely be identified in the future. In addition, for foliar diseases like dollar spot, the mode of action and critical compounds of *Trichoderma* may differ from that of soil and root pathogens.

*Trichoderma* induces a dynamic hormonal and metabolic crosstalk in plants, and the expression of auxin, JA/ethylene, and SA-related genes may overlap in plants, as determined by the *Trichoderma* strain and its inoculum amount, the plant species, the plant’s developmental stage, and the timing of the interaction. After studying 83 samples, our previous study concluded that *T. virens* 192-45 strains showed better inhibition of the *Clarireedia homoeocarpa* pathogen through the competition for space and nutrients in plate culture and the secretive non-volatile and volatile substrates [28]. Based on metabolome, transcriptome, and physiological analyses, this study aims to test the functional relevance of genes and metabolites whose expression is modulated during the interaction in the plant and the fungi in *Trichoderma*–plant interactions in the presence of *C. homoeocarpa*. Using the *Trichoderma* strain and its primary or secondary metabolites will be a powerful tool to establish the ecological roles of these signals.

## 2. Materials and Methods

### 2.1. Growth Condition of Plant and Fungi

The creeping bentgrass (*Agrostis stolonifera*, *As*) cultivar ‘Tiger’ was used. Dried seeds were sterilized with 75% ethanol for 3 min followed by 3% sodium hypochlorite for 10 min. Seeds were germinated on a substrate (peat soil: sand = 1:1) sterilized by high temperature and pressure in plastic pots (16.5 cm height with 16.5 cm diameter). The plants grew in the growth chamber under controlled conditions (300 µmol photons m^−2^ s^−1^ of photosynthetic photon flux density light; 75% of relative humidity; 25/20 °C of a continuous day/night temperature and a 12/12 h of day/night cycle, respectively). Grasses were grown about 20 days and mowed at the height of 5 cm before inoculation. The first experiment was carried out in November 2020, and the experiment was repeated in June 2021.

The pathogen strain of dollar spot was previously collected from *As* leaves infected by dollar spot, located at the putting green of Beijing Qinghe Bay Golf Course. Based on their morphology and the blast result of ITS sequence, it was identified as *C. homoeocarpa* (GenBank accession number: MW455464). Several *Trichoderma* strains were collected from grassland across Beijing and Jiangsu Province. Among them, *T. virens* strains 192-45 (GenBank accession number: MW455002) showed better pathogen inhibition through the antagonism experiments in plate culture. These fungi, as mentioned above, were maintained on potato dextrose agar (PDA).

### 2.2. Plant Inoculation Experiments and Sampling Point

*T. virens* strain 192-45 was used in these studies. The fungus was propagated on PDA in darkness for 7 days at 28 °C. The conidial suspension was obtained by washing the surface of the fungal colony with sterile distilled water. The inoculum was adjusted to 1 × 10^6^ spores, and 50 mL spore suspension was applied to 20-day-old *As* seedlings for 3 consecutive days. After 24 days of co-cultivation (colonization verification was detected by qPCR), plant growth was determined and pathogen infection assays were conducted.

For *As* infection assay, *C. homoeocarpa* were first prepared by pre-growing on the center of PDA plates for 5 days, allowing for the development of adequate mycelia. *C. homoeocarpa* mycelial suspension was obtained by scraping the colony’s surface with a scalpel and sterile distilled water. The *As* plants were sprayed and irrigated with 50 mL *C. homoeocarpa* mycelial suspension for 3 consecutive days.

Thus, all plants were divided into four groups: (a) well-watered without any treatment (CK); (b) *T. virens* pretreatment (Tv); (c) *T. virens* pretreatment and *C. homoeocarpa* infection (TC); and (d) only *C. homoeocarpa* infection (Ch). There were five replicate pots per treatment. For all treatments, leaf samples were observed for disease symptoms and collected at 7, 10, 13, 16, and 19 days postinoculation (dpi) of *C. homoeocarpa* for subsequent assay.

### 2.3. Determination of Plant Growth and Physiological Characteristics in Creeping Bentgrass

*Chlorophyll content*: Total chlorophyll content (Chl) was extracted by soaking 50 mg fresh leaves in 10 mL dimethyl sulfoxide in the dark for 72 h. Afterward, the homogenate was measured with a UV-visible spectrophotometer (MAPADA, Shanghai, China) at 663.2 nm, 646.8 nm, and 470 nm. Chlorophyll content was calculated by the formula of Wellburn [29].

*Phytohormone content*: To extract active phytohormones, including salicylic acid (SA), jasmonic acid (JA), indole acetic acid (IAA), and abscisic acid (ABA), *As* leaves (0.1 g) at 19 dpi were collected, weighed, immediately frozen in liquid nitrogen, and finely ground into powder. This was followed by extraction with 500 μL of modified Bieleski solvent (methanol/H_2_O, 80/20, *v*/*v*) at 4 °C for 12 h. The test samples were then analyzed on the HPLC-MS/MS system consisting of an Agilent 1290 series LC system (Agilent, Mississauga, ON, Canada) connected to AB Sciex QTrap^®^ 6500 mass spectrometers (AB Sciex, Concord, ON, Canada) located in Nanjing Ruiyuan Biotechnology Co., Ltd. (Nanjing, China). Each sample was measured with three replicates of all mixed leaves.

*Antioxidant assay*: A powder of approximately 0.1 g was suspended in 1.8 mL of extraction buffer (pH 7.8), which consists of 50 mM potassium phosphate, 1 mM ethylenediaminetetraacetic acid, and 1% polyvinylpyrrolidone. All mixtures were vortexed for 15 min and then centrifuged at 4 °C for 30 min at 13,000× *g*. The supernatant (about 1.5 mL) was collected for protein quality, enzyme assays, and MDA content. Protein concentration was quantified using the Bradford method [30]. The activities of antioxidant enzymes, including superoxide dismutase (SOD), catalase (CAT), peroxidase (POD), and ascorbate peroxidase (APX), were determined according to the method of Zhang and Kirkham [31].

*Malondialdehyde content*: Lipid peroxidation was estimated by measuring the content of malondialdehyde (MDA) with some modifications [32,33]. Briefly, a 0.5 mL aliquot of supernatant from enzyme extraction was mixed with 2 mL of 20% trichloroacetic acid containing 0.5% thiobarbituric acid. The mixture was heated at 95 °C for 30 min, cooled on ice, and then centrifuged at 10,000× *g* for 10 min. The absorbance was read at 532 and 600 nm. MDA concentration was calculated using an extinction coefficient of 155 mM^−1^·cm^−1^.

*Phenylalanine ammonia-lyase (PAL) activity*: PAL activity was determined according to the methods of Lister et al. and Aydaş et al. with some modifications [34,35]. The reaction consisted of 200 µL crude enzyme, 200 µL l-phenylalanine (80 mM), and 1.6 mL phosphate buffer (50 mM, pH 7.0). The mixture was incubated at 30 °C for 60 min in the water bath. After incubation, a 200-µL aliquot of 20% trichloroacetic acid (TCA) was added to terminate the reaction, and tubes were centrifuged at 5000× *g* for 5 min to pellet the denatured protein. PAL activity was determined from the yield of cinnamic acid, estimated by measuring A290 of the supernatant in 10 mm quartz cuvettes. The results were expressed as units per mg of protein.

*Polyphenol oxidase (PPO) activity*: PPO activity was determined according to the methods described by Naing et al. [36], with some modifications. The mix of 100 µL crude extract and phosphate buffer (50 mM, pH 6.4) and 1 mL catechol (20 mM) were incubated in a tube at 37 °C for 5 min (hydrothermal), respectively. Two tubes were immediately transferred to the cuvette. The polyphenol oxidation of PPO was then determined by the changes in absorbance at 420 nm with a UV spectrophotometer. The results were expressed as the changes in A/min per mg of protein.

*Statistical analysis*: The data of above physiological characteristic were statistically analyzed using GraphPad software (Beijing, China). Statistically significant differences were determined through univariate and multivariate analyses with Tukey tests and ANOVA model. Different letters were used to indicate means that differ significantly (*p* < 0.05).

### 2.4. Transcriptome Analysis of As Leaves and qRT-PCR Vertification

In the current study, leaves of all treatments at 19 dpi were collected for transcriptome analysis. In brief, total RNA was isolated from mature leaf tissues using TRIzol according to the manufacturer’s protocol. RNA-Seq was performed on three biological replicates of each sample. Oligo(dT)-attached magnetic beads were used to purify mRNA. The product was validated on the Agilent Technologies 2100 bioanalyzer (Agilent Technologies, Beijing, China) for quality control. The double-stranded PCR products from previous step were heated, denatured, and circularized by the splint oligo sequence to obtain the final cDNA library. The final library was amplified with phi29 to make DNA nanoball (DNB), which had more than 300 copies of one molecular; DNBs were loaded into the patterned nanoarray and single-end 50 base reads were generated on BGIseq500 platform (BGI-Shenzhen, China).

The raw data were filtered to the clean data using SOAPnuke (BGI-Shenzhen, China) software and clean reads were de novo assembled with Trinity software [37]. Then, TGICL were used to cluster the assembled transcripts to remove redundancy and map them to the Unigene library [38]. The quality of the assembled transcripts was assessed with BUSCO [39]. The annotation of unigenes was obtained by sequence alignment by searching for Nr, Nt, Swissprot, Pfam, KOG, and Kyoto Encyclopedia of Genes and Genomes (KEGG) public databases. Gene Ontology (GO) annotations and functional term mapping of transcript were performed using Blast2GO software and the eggNOG-mapper v2 database [40,41]. The KEGG pathway analysis was performed for mapping proteins onto known pathways [42]. All raw data and datasets were submitted to the China National Center for Bioinformation under the BioProject PRJCA010875.

For gene expression analysis, clean reads were aligned to Unigene by Bowtie2, the gene expression of each sample was calculated with fragments per kilobase of transcript per million mapped reads (FPKM) using RSEM software, and differential expression (DE) analysis of pairwise comparison (Tv vs. CK and TC vs. Ch) was performed by DEseq2 software under two conditions with false discovery rate ≤ 0.05 and Log2 fold change ≥2 [43]. The statistical enrichment of differential expression genes (DEGs) in KEGG pathways and GO term was performed using the ClusterProfiler R package [44]. The aforementioned transcriptome sequencing and differential analysis were completed by BGI Co., Ltd. As a way to investigate detailed functional categories and the expression level of DEGs, visualizations of KEGG classes and expression were also generated using R packages provided by ggplot2 and complexheatmap.

For the qRT-PCR verification, total RNAs were extracted from plant leaves of the same plot using TRIzol according to the manufacturer’s protocol, and three biological replicates were used. cDNA was obtained from total RNA with gDNA removal and the cDNA synthesis supermix kit (TransGen Biotech, Beijing, China). The qRT-PCR assays were performed using the SYBR Mix Kit (TransGen Biotech, Beijing, China) and were conducted on a Roche LightCycler 96 Sequence Detection System. The reference gene, *CdActin*, was used to normalize the expression levels of target genes. The expression levels were calculated using the 2^−ΔΔCt^ method. The qRT-PCR analysis was performed to validate the reliability of the RNA-Seq data and analyze the expression level of essential genes. qRT-PCR data were determined through univariate and multivariate analyses with Tukey tests and ANOVA model. Different letters were used to indicate means that differ significantly (*p* < 0.05).

### 2.5. Metabolome Analysis of Plant Leaves

*As* leaves of all treatments at 19 dpi were collected to perform untargeted metabolomics analysis. Five independent biological replicates were used. Collected samples were immediately frozen in liquid nitrogen, and 50 mg of tissues were weighed and extracted by directly adding 800 μL of precooled extraction reagent (MeOH: H_2_O (70:30, *v*/*v*, precooled at −20 °C)). After filtering, 600 μL of the supernatants were transferred to autosampler vials for UPLC-MS analysis. The sample analysis was performed on an ACQUITY UPLC 2D (Waters, MA, USA), coupled to a Q Exactive mass spectrometer (Thermo Fisher Scientific, Waltham, MA, USA) with a heated electrospray ionization (HESI) source in BGI Co., Ltd. The BGI’s metabolomics software package metaX and metabolome information analysis process was used for data preprocessing, statistical analysis, metabolite classification, and functional annotation. In this study, the value of variable importance in projection (VIP) of the first two principal components with partial least squares discriminant analysis (PLS-DA) model, combined with fold change (FC) ≥ 1.2 or ≤0.83 and Student’s *t*-test of univariate analysis (*p*-value < 0.05) were set as the filtering condition to choose metabolites with different abundance (DEM). The volcano plot and heatmap of DEM were obtained in R using the packages of ggplot2 and ComplexHeatmap [45,46].

### 2.6. Untargeted Metabolic Analysis and Efficacy Experiment of Respective Metabolites of T. virens

For exploring the kinds and abundance of metabolites produced by *Trichoderma*, the single colony of *T. virens* 192-45 with five biological repeats was also collected to perform untargeted metabolomics analysis based on the above-mentioned method. If there is a missing value in the abundance of a metabolite in a repeated strain, the metabolite will be filtered out. After this strict filtering, 446 metabolites were obtained and sorted according to their abundance. Similarly, the classification and functional annotation of metabolites were defined by BGI’s metabolomics software. Based on the classification and annotation, the bar chart of metabolites class and heatmap of metabolites with top 100 abundance were generated in R using the packages of ggplot2 and ComplexHeatmap [45,46].

To investigate the disease control efficacy of citric, linolenic, and fumaric acid of *T. virens*-emitted respective metabolites, in vitro experiments were carried out, using the pure compounds purchased from Sigma (Sigma, St. Louis, MO, USA). The individual pure compounds of respective metabolites were prepared by dissolving in a 5% methanol or ethanol solution. The activity of respective metabolites was tested against the *C. homoeocarpa* pathogen with the agar diffusion test. A plug (5 mm) of mycelium from an actively growing culture was placed in the center of the PDA plate, and 10 µL of metabolite solution at different concentrations (0, 50, and 100 mg/mL) was placed at a distance of 25 mm. The plates were incubated at 30 °C for 3 days. The growth inhibition was assessed by the photo of the inhibition halos compared with the control (5% methanol or ethanol).

## 3. Result

### 3.1. Effect of T. virens on Plant Growth and Physiological Characteristics in Creeping Bentgrass against C. homoeocarpa

The pot experiment assessed the physiological and pathological processes of different inoculations to gather information about the overall biostimulant effect of *T. virens*. We found that the pretreatment of *T. virens* effectively alleviates the development of dollar spot caused by *C. homoeocarpa* inoculation (Figure 1a). Furthermore, this positive effect of *T. virens* is mainly reflected in the chlorophyll degradation and MDA content accumulation (Figure 1b,c).

Moreover, plants treated with *T. virens* and *C. homoeocarpa* have different effects on the activity of antioxidants. The activities of SOD and POD in *As* plants were significantly stimulated by inoculation of *C. homoeocarpa*, irrespective of the presence of *T. virens* (Figure 1d,f). Meanwhile, compared to the control, the co-treatment of *T. virens* and *C. homoeocarpa* induced strong APX activities in the plants (*p* < 0.05, Figure 1g). Similar results were also seen in PAL activity (Figure 1h). In particular, the greatest plant-resistance effect in *T. virens* was observed when the PPO activity significantly increased in the co-treatment group of *T. virens* and *C. homoeocarpa* compared to the individual treatment (Figure 1j). In addition, we repeated the whole experiment to confirm the effect of *T. virens* on the plant growth and resistance of *As* plants.

Quantitative analysis of endogenous hormone of plant leaves with different treatments would facilitate understanding of the potential mechanism of *T. virens*. As shown, upon the *T. virens* treatment, the leaves accumulated relatively high levels of IAA content compared with the control plant (Figure 2a). However, the changes in ABA content in the *T. virens*-pretreated plant were less pronounced (Figure 2b). Meanwhile, after pretreatment of *T. virens*, the ABA, SA, and JA content were significantly higher (*p* < 0.05) in the TC group than in the Ch group (Figure 2c,d).

### 3.2. Transcriptional Expression Analysis of As Leaves’ Response to T. virens Colonization and Infection by C. homoeocarpa

To study the potential roles of *T. virens* in defense against *C. homoeocarpa*, we evaluated transcriptional changes in *As* leaves after colonization of *T. virens* and 19 dpi of infection of *C. homoeocarpa*. A total of 76.56 Gb data was obtained. After assembly and redundancy removal, 95,299 unigenes were obtained, with total length, average length, N50, and GC content of 91,516,445 bp, 960 bp, 1382 bp, and 54.57%, respectively. A total of 82,289 unigenes were successfully annotated to at least one database.

As shown in Figure 3a, pairwise comparisons across different treatments revealed a range of differentially expressed genes (DEGs). Of these, 77 and 127 transcripts were up- and downregulated in the *T. virens* inoculation (Tv) versus control (CK). The KEGG term of these DEGs was associated with the metabolism of primary and secondary, energy metabolism, and genetic information processing (Appendix A). For instance, transcription factor IIE (TFIIE) factors and LRR receptor-like serine/threonine protein kinase (LRR), known for functions in plant–microbe interactions, were upregulated in the Tv group (Figure 3b) [47,48]. The expression pattern of other receptor-like kinases, like proline-rich receptor-like protein kinase (PERK2, 8), appears to be as diverse as its function. In addition, a member of aldehyde dehydrogenase family 2 (ALDH2), two cinnamyl alcohol dehydrogenase (CAD6, 7), and one multiprotein bridging factor 1 (MBF1) are likely involved in various abiotic stress and biotic stimuli [49,50,51]. Moreover, two vegetative cell wall proteins (GP1) and two fructan exohydrolases (6-FEH) are structurally related to cell wall expansins [52]. There was also an upregulation of a set of genes involved in terpenoid metabolism and stress resistance metabolism, such as premnaspirodiene oxygenase (CYP71D55), 9-cis-epoxycarotenoid dioxygenase (NCED1), ethylene-responsive transcription factor RAP2-7-like (AP2/ERF), along with three genes encoding dormancy/auxin associated proteins (DRM1/ARP). However, we found a group of DEGs encoded heat shock protein was downregulated in Tv compared to the CK.

We then focused on the *C. homoeocarpa* infected leaves with or without *T. virens* treatment. In total, 311 and 2521 transcripts had up- and downregulated expressions in the Ch group relative to those in the TC group, respectively (Figure 3a). Besides, the DEG number of TC vs. Ch (2832) was much more numerous than in the comparison of TC vs. Tv (233), and there are more upregulated genes in the Ch group. Although 2521 upregulated genes were found in the Ch group (Appendix A), a set of disease resistance DEGs were only upregulated in the TC group (Figure 3c). For example, four putative disease resistance proteins (RGA2) and two calcium-dependent protein kinase 4 (CPK4), commonly known for positive functions in plant immune responses and pathogen invasion, were found in the TC group [53]. Moreover, we found a group of upregulated transcriptional factors associated with plant defense against the pathogen infection under the presence of *T. virens*. For instance, R2R3-type MYB transcription factors and enhancer of TRY and CPC3 (MYB-ETC3), CAPRICE (MYB-CPC), in addition to heat stress transcription factor A-2c (HsfA2c) and MADS-box transcription factor 4 (MADS4) were identified [54]. Several receptor-like kinases (RLKs) encoding genes associated with plant defense response, were also present in this list, such as proline-rich receptor-like protein kinase (PERK1), FERONIA (FER), putative F-box/LRR-repeat protein, MLO-like protein 1 (Mlo1), Casein kinase 1-like protein 3 (CKL3), and cysteine-rich receptor-like protein kinase 40 (CRK40) [55,56]. However, much more genes encoded as leucine-rich repeat extension-like protein 5 (LRX5) were downregulated in the TC group. Moreover, a group of DEGs associated with plant immunity and hormone metabolism of basal resistance was also upregulated in the TC group, such as HSPRO2, which encodes a nematode resistance protein [57], and CRK11 (cytokinin dehydrogenase 11), related to disease resistance [58]. The expression of four candidate genes (*ARP*, *NCED*, *CPK4*, *HSPRO2*) was also determined by qRT-PCR, which closely matched RNA Seq data over most samples (Appendix A).

### 3.3. Metabolome Analysis of Resistance-Related Pathway Changes in As Leaves

To further characterize *As* responses to *T. virens* and *C. homoeocarpa*, leaves were extracted from the Tv, TC, Ch, and CK groups and profiled using a non-targeted metabolomics approach. The UPLC-MS method resulted in 14,854 molecular features with unique *m*/*z* values and retention time, where 1421 compounds could be predicted based on the reliability level of metabolic identification.

There are 170 metabolites with differential abundance (DEM) between the Tv and CK groups, among which 101 metabolites were decreased, and 69 metabolites were increased in *As* pre-contacted with *T. virens* (Appendix A). Filtering these DEMs by the VIP of PLS-DA and classification after combing HMDB and KEGG databases, 66 metabolites were found to be involved in both primary and secondary metabolic categories, such as alcohols (i.e., shikimic acid), amines and amino acid (i.e., spermidine and *N*-acetyl-l-phenylalanine), endogenous metabolites (i.e., *N*-acetylornithine), steroids, and terpenoids (i.e., isosteviol, citral) (Figure 4a). Apart from that, several interesting metabolite analogues seem to be induced by *T. virens*, such as bilirubins, the animal hormone epinephrine, and the drug pravastatin. In addition, 52 compounds were still classified in publicly available databases.

Across the comparison analysis between the TC and Ch group, 71 metabolites were increased, and 69 metabolites were decreased in As co-contacted with *T. virens* and *C. homoeocarpa* (Appendix A). Similarity, 41 DEMs filtered by the VIP value of the TC and Ch comparison with precise classification were identified (Figure 4b), mainly involved in L-phenylalanine metabolism, fatty acyls, flavonoids, and steroids. Of these, the metabolites of l-phenylalanine metabolism deserve more attention. The branch metabolites of l-phenylalanine degradation, such as phenylacetylglycine, *N*-acetyl-l-phenylalanine, *N*-acetyl-l-tyrosine, and hydroxyphenyllactic acid, presented lower abundance in the TC group. However, the biosynthetic products of l-phenylalanine that were increased in the TC group, such as phenylpropanoids (i.e., rosarin), contribute to plant response to abiotic and biotic stresses [59]. Except for the metabolites related to l-phenylalanine metabolism that appeared in the TC group, *N*-acetylornithine of spermidine biosynthesis was also accumulated in the TC group. Furthermore, several DEMs increased in the TC group could be matched to the family of fatty acyls, flavonoids, and steroids (Figure 5). Examples include stearic acid, associated with the induction of constitutive defense signal [60]; gypenoside, associated with synergistic antifungal effects [61]; and mulberrin, related to antimicrobial activity.

### 3.4. Untargeted Metabolic Analysis and Efficacy of Biocontrol-Related Metabolites of T. virens

In the case of non-volatile metabolites, our previous work showed that the *T. virens* 192-45 strain reduced the growth of *C. homoeocarpa* by 89.47% in a PDA medium with removed cellophane and *T. virens* growing for 7 days [28]. Thus, for the detection of the metabolites of *T. virens* 192-45 strain, 445 compounds were obtained via UPLC-MS analysis. Of these, 176 compounds possessed unambiguous classification and were enriched in organic acid, derivatives, fatty acyls, benzenoids, and steroids and products.

To gain insight into these metabolites produced by *T. virens*, the metabolites with top 100 abundance were selected by their abundance in the peak area (Appendix A and Figure 5). Using UPLC-MS analysis, a set of the metabolites with higher abundance are concentrated in organic acids, organic nitrogen, organoheterocyclic compounds, phenylpropanoids, and peptides, which may be related to pathogen inhibition and induced resistance. For example, malic acid and citric acids are fungicides or insecticides. The discovery of six metabolites from *T. virens* is of particular concern, since they all appear to be involved in linolenic and oleic acid biosynthesis or their precursors and intermediates, jasmonate biosynthesis, and inducible defenses. These were 13(S)-hydroxy octadecatrienoic acid (13-HOTrE), 12-oxo-phytodienoic acid, 13S-hydroperoxy-linolenic acid (13S-HPOTrE), 9,10-epoxyoctadecenoic acid (9,10-EpOME), 4-dodecylbenzenesulfonic acid, and oleic acid (Figure 5). In the present study, the agar diffusion test of citric acid and linolenic acid at 100 mg/mL inhibited the growth of *C. homoeocarpa* (Appendix A). In addition, several secondary metabolites related to disease resistance were obtained, such as hydroxycinnamic acids, ipriflavone, L-sorbose, allopurinol, oxipurinol, 16-hydroxyhexadecanoic acid, and asparagine [62,63,64,65].

## 4. Discussion

The use of *Trichoderma*-based bioinoculants in agriculture is increasing, with several hundred products available worldwide. Studies evaluating the mechanism of *Trichoderma*–plant interaction have been much more numerous. However, the crosstalk of metabolites and signal in the plant and *Trichoderma* is dynamic [66]. The changes of respective genes and metabolites related to plant growth and disease resistance depend on the *Trichoderma* strains and the plant species. In this study, we investigated previously unknown mechanisms underlying *As* interaction with *T. virens* against *C. homoeocarpa* from multiple angles. We investigated transcriptional alterations of leaves treated with *T. virens* and/or *C. homoeocarpa*. Untargeted metabolomes were profiled to elucidate the chemical nature of *T. virens* serving as a critical trigger for the metabolome of *As* leaves. The mechanism of *T. virens*–*As* interaction can be viewed in two respects: plant growth promotion and plant defense response, both discussed below (Figure 6):

### 4.1. Effect of T. virens on As Plant Growth Regulation

It has been known for decades that *Trichoderma* can impact plant growth on axenic and soil systems. Plants may benefit from *Trichoderma*’s ability to induce development changes through different mechanisms, such as the production of phytohormones and other secondary metabolites [66].

Firstly, the potential of plant-associated *Trichoderma* to produce free IAA and intermediates or precursors represents a means to influence the endogenous auxin pool of the host and induce IAA-responsive protein [67]. Plants may benefit from *Trichoderma*’s ability to induce development changes through different mechanisms. In this study, various carbohydrates and nucleic acids decreased in the Tv group (Figure 4a), indicating that the plants treated by *T. virens* had more vigorous life activities and consumed more carbohydrates. It is worth noting that this study identified Indole-3-acrylic acid and tryptophan from *T. virens* hypha (Figure 5). Indole-3-acrylic acid is generated from tryptophan and reported as an auxin growth regulator [68]. Additionally, L-tryptophan has been found to stimulate the synthesis of auxins in the rhizosphere and trigger plant growth [69]. A transcriptional profile of *As* leaves treated with *T. virens* revealed the differential expression of 204 genes, including two components of dormancy-associated gene 1/auxin-repressed protein (DRM1/ARP) genes (Figure 3a,b). DRM1/ARPs are responsive to auxin that regulates plant growth and development [70]. In this study, the accumulation of IAA content was also higher in the *As* plants only inoculated with *T. virens*. In addition, it is apparent that *As* plants inoculated with *Trichoderma* grow better (Figure 1a). In summary, these results suggest that *T. virens* produces IAA and related compounds in order to promote plant growth and auxin biosynthesis through regulating DRM1/ARP expression (Figure 6).

Secondly, spermidine, as the main component of polyamines, is essential for cell division and proliferation and is implicated in diverse plant growth and development, in addition to stress tolerance [16,71]. Regarding hormone signaling pathways, spermidine positively regulated auxin and cytokinin signaling genes [72]. Firstly, the Dl-arginine, potentially as a spermidine stimulus, was obtained from *T. virens* hypha. Furthermore, spermidine and its precursor *N*-acetylornithine are increased in *As* leaves treated by *T. virens* (Figure 4a). Secondly, a group of DEGs was upregulated in Tv-treated *As* plants with functions in auxin, ethylene, and abscisic acid biosynthesis and signaling pathways, including DRM1/ARP-, AP2/ERF-, and NCED1-encoding genes. Transcriptome and metabolome data indicate a potential of higher spermidine to induce hormone-related genes promoting plant growth and stress tolerance (Figure 6).

Thirdly, the global transcriptional changes of *As* genes encoding receptor kinases (i.e., LRR, PERKs) and cell wall biogenesis (i.e., GP1 and 6-FEH) are upregulated in *As* leaves pre-contact with *T. virens*, which are all well known to be the environmental stimulus [47,48,52] Additionally, via untargeted metabolome analysis, several interesting substrates were obtained from the Tv-treated *As* leaves, including bilirubin, epinephrine, and isosteviol (Figure 4a). Bilirubin is involved in the biosynthesis of various plant secondary metabolites, and foliar spraying positively affects maize growth and yield [73]. Epinephrine can stimulate somatic embryogenesis in leaf cultures [74]; however, knowledge about the function and regulation of epinephrine in plants is still limited. Isosteviol shows similarities with gibberellins in their matrix structure and could positively promote wheat seeds’ growth indexes [75].

Although some substrates are low in content, they may play a key role in promoting plant growth. There are, for example, indole-3-acrylic acid and l-tryptophan, associated with auxin synthesis and regulation; glutamine and agmatine, associated with arginine and sustaining growth; and succinic acid and fumaric acid which, when fed to plant TCA cycle and promote plant growth and trigonelline, a diverse plant regulator. Of course, many metabolites with relatively low concentration and no precise classification still need further research. Overall, the current results strongly suggested that the ‘pre-defense’ was activated by the approaching *T. virens* from two aspects of gene expression and metabolic changes.

### 4.2. Induction of Defense Response by T. virens on As Plant

There were more DEGs in the comparison of TC with Ch than in the comparison of TC with Tv, and there were more upregulated genes in the individual treatment of *C. homoeocarpa* (Figure 3a). The results indicated that reducing sensitivity to the pathogen can be attributed to pre-contact with *T. virens*, and can act as the second barrier to protect plants challenged with *C. homoeocarpa* through various mechanisms (Figure 6).

In this study, the abundance of phenylacetylglycine and *N*-acetyl-l-phenylalanine decreased in the Tv group and further reduced in the TC group. In addition, hydroxyphenyllactic acid is the newly added component of the metabolites branch of l-phenylalanine degradation (Figure 4b). The reduction of these metabolites results in the accumulation of l-phenylalanine. Moreover, l-phenylalanine is the precursor of cinnamyl alcohol glycosides (collectively named rosarins) through the catalytic reaction of phenylalanine ammonia-lyase (PAL), cinnamyl-CoA reductase (CAD) enzyme, and others [76,77]. Furthermore, rosarins may possess immuno-stimulant properties [78]. These findings are consistent with the enhancement of PAL activity in *As* plants treated with *T. virens* (Figure 1h) and the abundance of l-phenylalanine in *T. virens* (Figure 5). These results indicate that *T. virens* could mediate the pathway of L-phenylalanine metabolism to resist diseases (Figure 6).

Organic acids, including oleic and linolenic acid detected in *Trichoderma*, were reported to possess antifungal activity and influence pathogen metabolism [79,80]. The present study also found that citric acid and linolenic acid inhibited the growth of *C. homoeocarpa* on agar diffusion plates (Appendix A). In the meantime, precursors and intermediates of oleic acids and linolenic acid also play selective roles in regulating SA-related or JA-related defense resistance, such as 13-HOTrE, 13S-HPOTrE, OPDA, and 9,10-EpOME. Signaling induced upon a reduction in oleic acid (18:1) levels simultaneously upregulates salicylic acid (SA)-mediated responses and inhibits jasmonic acid (JA)-inducible defenses [60,81]. Furthermore, salicylic acid signaling may in part be mediated by 13-HOTrE in barley [82]. On the other hand, OPDA is the first cyclic compound in jasmonate biosynthesis, and 13S-HPOTrE is a precursor of jasmonic acid [82,83]. Furthermore, cytochrome P450-derived linolenic acid metabolites, 9,10-EpOME, have been studied for their association with various disease states and biological functions [84]. This is consistent with the detection of oleic acid and intermediates from *T. virens* (Figure 5) and the increase in stearic acid in *As* leaves treated with *T. virens* and *C. homoeocarpa* (Figure 4b). Minichiello et al. reported that oleic and linoleic acids substantially activate oat leaf Ca^2+^-dependent protein kinase under high temperature [85]. In this study, the expression of several genes and the abundance of stearic acid were only increased in the group of *As* plant co-treated with *T. virens* and *C. homoeocarpa* infection, such as the CPK4, MADS4, and HSPRO2 gene (Figure 3c), that are related to the defensive response in plants [53,86,87]. In line with this, the content of SA and JA were increased in *As* plants co-treated with *T. virens* and *C. homoeocarpa* infection, compared to those only treated with pathogen infection. Collectively, oleic and linolenic acid or their precursors and intermediates secreted by *T. virens* tend to be major compounds that influence the accumulation of stearic acid and SA, and induced disease defense genes (Figure 6).

In conclusion, our current data reveal that *As* leaves dynamically respond to the presence of *T. virens* with or without *C. homoeocarpa* infection (Figure 6). During the absence of pathogens, plant growth and physiological performance is enhanced by the production of bioactive metabolites, including hormones and spermidine, which are prepared for the ensuing pathogen attack by using ‘pre-defense’ (Figure 6). Upon invasion by *C. homoeocarpa*, the SA- and/or JA-dependent defense system of *Trichoderma*–plant interactions was induced by *T. virens* and its emitted metabolites. Using synthetic analysis, we can infer that secondary and primary metabolites, such as oleic and linolenic acids and their intermediates, have functions we have not yet discovered, which could be used for exogenous application to control diseases in the future. Subsequently, several genes related to plant growth promotion and disease resistance induced by *T. virens* were found in *As* plants. We anticipate a combination of genetic engineering and breeding approaches targeting these genes could make creeping bentgrass more resistant to soil pathogens, with an ultimate goal of improving crop productivity.

## Figures and Tables

**Figure 1 jof-08-01186-f001:**
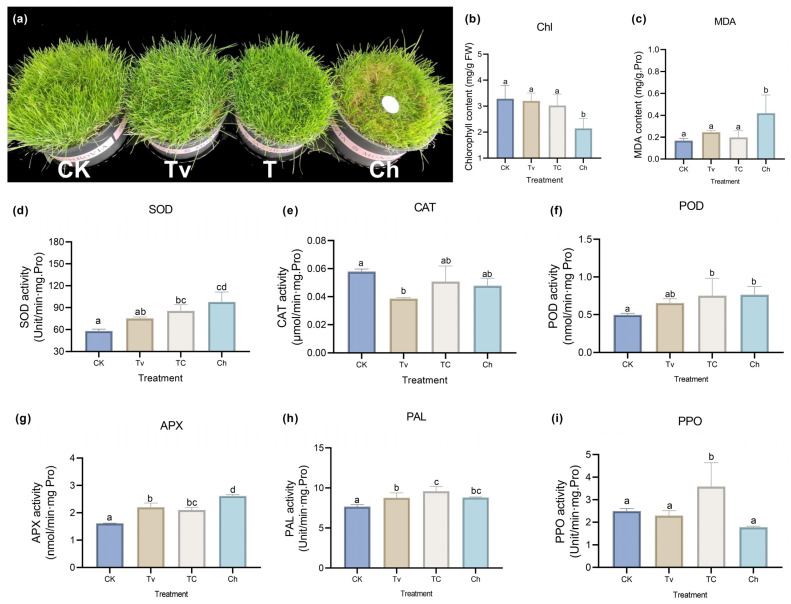
Physiological performance of creeping bentgrass after *T. virens* treatment and/or inoculation of *C. homoeocarpa* at 19 dpi. CK, plants with well-watered without any treatment; Tv, plants with the pretreatment of *T. virens*; TC, plants treated with T. virens pretreatment and C. homoeocarpa infection; Ch, plants treated with C. homoeocarpa infection (the below is the same). (**a**) Photos of creeping bentgrass after various treatments in pots. (**b**,**c**) Chlorophyll and MDA content. (**d**–**g**) The activity of antioxidants: SOD, POD, CAT, APX. (**h**,**i**) The activity of pathogen-related enzyme: PAL, PPO. Different letters aboved were used to indicate means that differ significantly (*p* < 0.05).

**Figure 2 jof-08-01186-f002:**
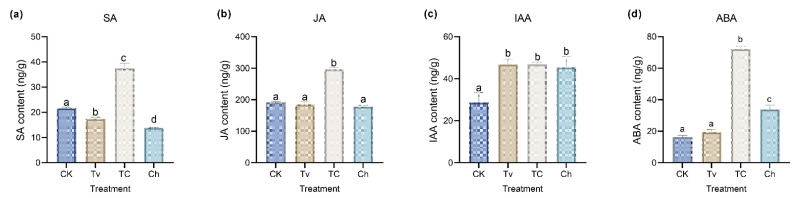
Phytohormone content of creeping bentgrass with *T. virens* treatment and/or inoculation of *C. homoeocarpa* at 19 dpi. (**a**) IAA. (**b**) ABA. (**c**) SA. (**d**) JA. Different letters aboved were used to indicate means that differ significantly (*p* < 0.05).

**Figure 3 jof-08-01186-f003:**
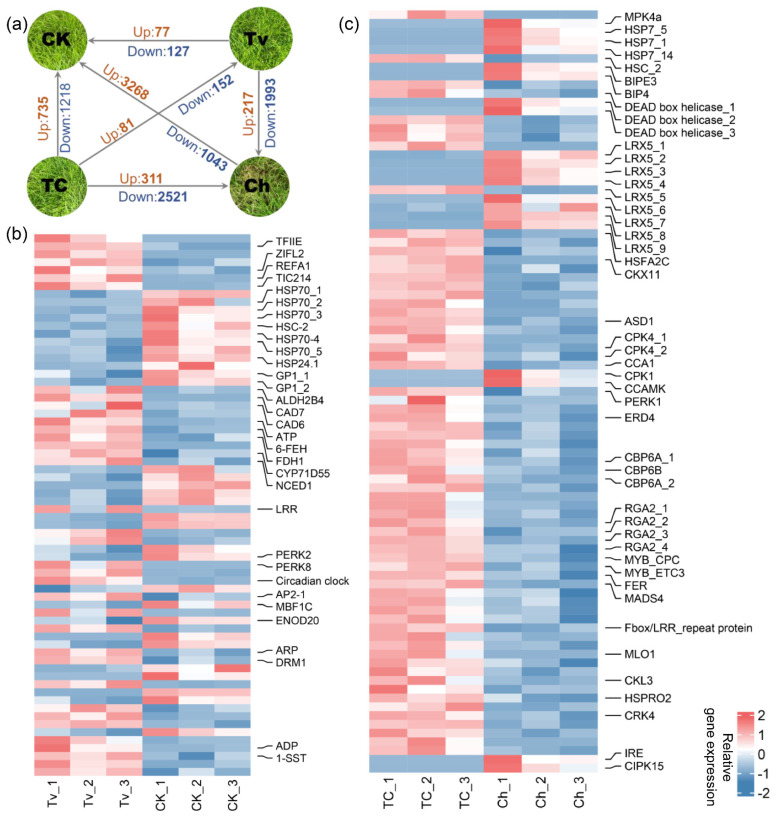
Transcriptome analysis of *As* leaves after colonization of *T. virens* and infection by *C. homoeocarpa* at 19 dpi. (**a**) Schematic overview of the DEG comparison with condition of |log2FC| ≥ 1 & Padj < 0.05 between different treatments. (**b**) Expression of DEGs (filtered with |log2FC| ≥ 2 & Padj < 0.01) between T. virens colonization (Tv) and control (CK). (**c**) Expression of DEGs upregulated in the TC group and their similar homologs (filtered with |log2FC| ≥ 2 & Padj < 0.01).

**Figure 4 jof-08-01186-f004:**
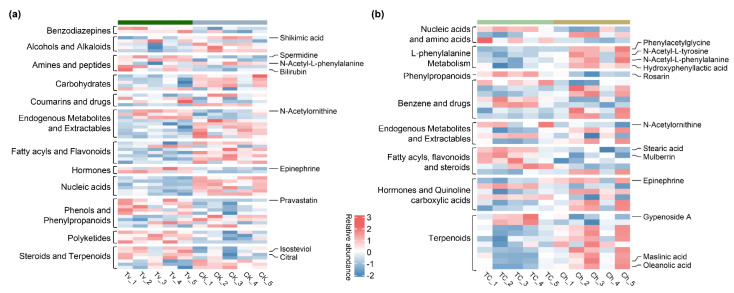
Untargeted metabolic profiling using the *As* leaves with *T. virens* colonization and *C. homoeocarpa* infection. (**a**) Heatmap of key metabolites with significantly different abundance (|log2FC|> 1.2, *p*-value < 0.05 & VIP > 1) in the plants between Tv and CK group (**b**) Pairwise comparisons across inoculation with *T. virens* and *C. homoecarpa* (TC) and *C. homoeocarpa*-only inoculation (Ch) with the same settings as (**a**).

**Figure 5 jof-08-01186-f005:**
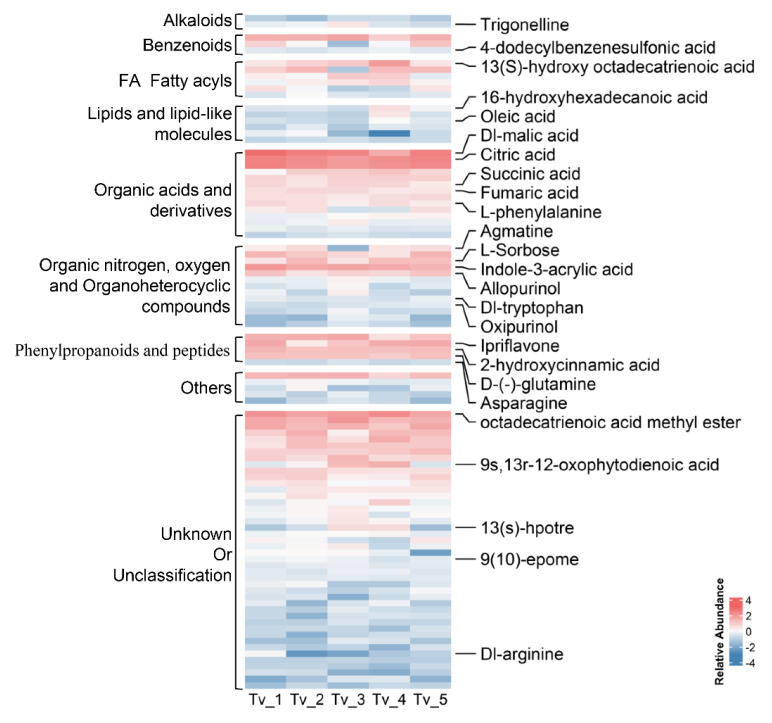
Untargeted metabolic profiling of *T. virens* 192-45 with the top 100 metabolites of high abundance from peak area.

**Figure 6 jof-08-01186-f006:**
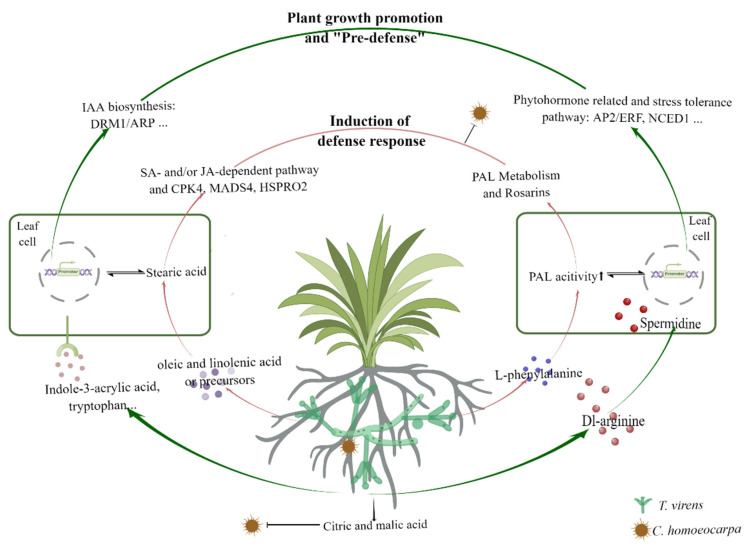
Overview of dual mechanism induced by *T. virens* under the *C. homoeocarpa* attack.

## Data Availability

The datasets presented in this study are available in Tables, Figures, Appendix A. The accession number(s) used in this study can be found in the article or Appendix A.

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
