# Peer review of "New Insights into the Mechanism of Trichoderma virens-Induced Developmental Effects on Agrostis stolonifera Disease Resistance against Dollar Spot Infection"

_jof, 2022, doi:10.3390/jof8111186_

Round 1

Reviewer 1 Report

accept

Author Response

Dear reviewer,

Thank you for your consideration.

Best regards,

Lu Gan

Reviewer 2 Report

dear authors, 

In introduction section, you write about Trichoderma too much, while you didn't indicate the economic importance of plant you used, you should add paragraph indicating this importance. 

In material and methods section,

the paragraph start with line 131 should be transferred after the paragraph start with line no 147.

in paragraph start with line no 207, you didn't indicate how you prepare the pure compounds before you used in this experiment, please clarify.  

In results section, panels a, b, d, e in figure 4 as well as panel a in Figure 5 need to resize to be seen. 

Author Response

Dear Reviewer,

Thank you for your valuable review and suggestions about our manuscript entitled “New Insights into the Mechanism of Trichoderma virens-induced Developmental effects on Agrostis stolonifera Desease Resistance Against Dollar Spot Infection” (ID: 1911343). Those comments are all valuable and important for improving the quality of our work. We have studied the comments carefully and would like to revise them (Note: The line numbers in all answers are modified positions). We have addressed the comments or response it as followed:

  1. In introduction section, you write about Trichoderma too much, while you didn't indicate the economic importance of plant you used, you should add paragraph indicating this importance.

Answer: Good point. We have added some related statements about the value of Agrostis stolonifera plants with two references at the Ln 32-36 in the Introduction section.

  1. In material and methods section, the paragraph starts with line 131 should be transferred after the paragraph start with line no 147. In paragraph start with line no 207, you didn't indicate how you prepare the pure compounds before you used in this experiment, please clarify.

Answer: This is a very good advice. We have played around with the order of the paragraph (Ln 154-160), modify some subtitles (Ln 130,182,226), and added the statement about the pure compounds (Ln 255-256) in the Methods.

  1. In results section, panels a, b, d, e in figure 4 as well as panel a in Figure 5 need to resize to be seen.

Answer: Indeed. To perform better, we would like to remove Figure 4a, d and Figure 5a to supplementary materials, and enlarge the text size in the two figures.

Reviewer 3 Report

The authors carried out a pot experiment to test the effects of a Trichoderma species on various aspects of creeping bentgrass (Agrostis stolonifera) physiology and gene expression, in the presence of a fungal pathogen. I think the results are interesting and of significance, but unfortunately the paper in its current state is not publishable.

My major concerns are the almost complete lack of detail in the ‘omics methods sections, particularly with regards to the analysis and the lack of detail regarding statistical analyses in general. Further, discussion points are interspersed within the results section and there are many sentences which defy interpretation, especially in the discussion section.

There are few details of the statistical tests used, and where given it is in garbled English. As far as I can work out the experimental design was 4x treatments with 5 reps of each treatment, 5x repeated measures sample points, 2x whole experiment replicated. I would need to know what statistical test and what model was fit for each analysis – this is especially important as it seems that results from a only a single (19DPI) measurement point (I’m aware that the ‘omics and qPCR analyses used only this sampling point). I’d also need to know how false discovery rate was controlled, possibly Tukey tests for enzymes and phytohormones, but there are no details for the ‘omics data. There are no details on the statistics used for the enrichment analyses – in fact the presented results for RNA-seq (fig S2.) don’t even look like an enrichment analysis. The only time that it’s clear what statistical analysis has been done is for the qPCR data, a Student t-test, and I’d suggest this is not the best method (ANOVA model + post-hoc test would be my choice). Finally, why were only 3 samples per treatment used for the ‘omics analyses and how were these selected.

The RNA-seq methods are thin in the extreme and need expanding on. Do you have a draft genome for As; a de-novo transcriptome assembly of a tetraploid species is not trivial. I really need a good description of the pipeline used. Also there are no results on the size or general quality assessment of the transcriptome. You need to give details of which software package was used for the DE analysis, and for the enrichment analysis (could just use a Fisher Exact test).  

The general structure for the methods section is fine, but each (relevant) subheading should have a few lines describing the statistical tests used. Then, the subheadings for the results section should use the same headings from the methods section. Avoid combining results from multiple analyses into a single section even if there are only a few words to say; I found it hard to work out which sections belong to which method. There’s also a lot of discussion in the results section which needs to be moved  - just concentrate on what you found. There’s no real discussion (though the methods suggest you measured) of grass performance – was there no discernible difference between the control and the 2 trich treatments?

The discussion at the moment is difficult to follow in places and needs editing, this is mostly just language usage. Finally, I think the final sentence needs to be toned down – the research is a long way from suggesting genetic manipulation.

Consider splitting some of the figures up to improve the text sizing, and especially get rid of any subplots if they’re not discussed in the ms anywhere.

Below is a line by line breakdown of suggested changes:

Lines 46-47 REF

Lines 48 – 49  “Also, there are few reports about the application of Trichoderma on turfgrass and pasture diseases.” This sentence needs to be moved, probably to the final paragraph of the introduction.

Line 152 “A Trichoderma ghanense strain GCPL175 has a more extraordinary ability to restrain Lepiota strain 1506” This would be better said as “Trichoderma ghanense strain GCPL175 has an extraordinary restraining effect on Lepiota strain 1506”

Line 64 – 65 “Although them,” This is not an idiomatic phrase and has no meaning, I’s suggest just deleting.

Line 67 delete “the metabolites of Trichoderma”

Line 84 “research data is under review”, change to submitted, or unpublished

Line 85 “our study aimed” change to (->) “this study aims”

Line 91 “fungal” -> “fungi” (and plant shouldn’t be capitalised)

Line 94 “on the sterilized” -> “on sterilized”,

Line 94  How was the substrate sterilised?

Line 101 “the As leaves” -> “As leaves”

Line 103 -103 “It was identified as C. homoeocarpa” How? Please add how it was identified.

Line 112 “applied for 20-day-old” -> “applied to 20-day-old”

Line 114 typo “pathogen infections were conducted” should probably be “pathogen infection assays were conducted

Line 118 “irrigated by” -> irrigated with

Line 137 Please check the units for extinction coefficient

Lines 94/135 (and others) Please ensure units are consistent min/minutes etc.

Line 142  is this HPLC-MS/MS,  LC-MS, LC-MS/MS  or UPLC-MS?  All 4 variants are used in the ms – please be consistent or give full methods if different instruments were used.

Line 151 – 152 “Protein concentration was quantified using the” incomplete sentence

Line 182 – 185 “In this study, the value of variable importance in projection (VIP) of the first two principal components in multivariate statistical analysis and univariate analysis, combined with fold-change(FC)≥1.2 or ≤0.83 and Student's t-test of univariate analysis (p-value<0.05) to choose differentially expressed metabolites” – this makes no sense please reword

Line 186 – 188 How was the enrichment analysis conducted?

Line 185 and throughout manuscript. Metabolites are not expressed. Please choose a different acronym – differential metabolite abundance? On the same not change up/down regulated for metabolites to increased/decreased abundance.

Line 192 – 193 “And cDNA libraries were constructed as previously described and sequenced by DNBseq”  this is an incomplete sentence and previously described by who and where? Please reword. General point, don’t start English sentences with “and” in scientific publication.

Section 2.6 Far more details on sample preparation is needed here. Also, for DNBseq was this SE/PE and length of inserts? How was transcriptome quality assessed? How was the annotation done – sequence alignment using what tools?

Line 194 – “reads were aligned with Bowtie software” Aligned to what?  Is there a genome available for As and was the denovo transcriptome assembly genome guided???  Please give all steps of how you’ve made a a denovo transcriptome for an allotetraploid – this is not a trivial undertaking. Also what proportion of reads aligned back onto the transcriptome. Further, the results don’t contain any details of the transcriptome – an absolute minimum would be the number of transcripts.

Line 197 – 200 RSEM is used for pseudo alignment, how was the differential analysis done?

Line 201 adjusted p-value – please add the method used for multiple testing adjustment

Line 202 annotation belongs to the de-novo transcriptome section, no?

Line 204 – 205 how was enrichment analysis performed and what statistical test/threshold was used for reporting

Line 220 “three biological replicated were used” -> “three biological replicates were used”. Do you mean technical replicates here, otherwise what happened to the other 2 biological replicates? This has statistical implications so is important to be precise. Also, which sampling times were used. The presented results (fig s3) has only a single time-point on it…

Line 220 – 221 “Total RNA was treated with gDNA Removal and used for the cDNA synthesis supermix kit” This doesn’t make sense. Please reword.

Line 229 – 232 “Statistically significant differences in plant traits were determined through univariate and multivariate analyses with Tukey tests” More details needed. Also given the factorial experimental design, some form of ANOVA + post-hoc test would be better to analyse the qPCR results than multiple t-tests.

Line 245 “induced plants' strong APX activities” -> “induced strong APX activity in the plants”

Line246 Figure 5h -> Figure 1h

Line 246 – 247 sentence beginning with “Remarkably”. This needs rewording as the meaning is unclear.

Line 249 – 250 “These results from these experiments indicate that T. virens played a crucial role in modulating the response and resistance of As against C. homoeocarpa (Figure 1, S1)” This is a discussion point not a result.

Line 250 – 252 “we repeated the whole experiment to confirm the effect of T. virens on the plant growth and resistance of As plants” and did it confirm this?? Either present the results for the repeated experiment, or better still combine the two experiments into a single analysis.

Line 258 “ T. virens of pretreatment,” typo please correct

Line 258 “ABA, SA, and JA content were significantly higher” please give sig values. Personally I like to see the test statistic and df (if appropriate) quoted as well, though journal style guide might not insist on this.

WERE’S THE DESCRIPTION/DETAILS OF THE DE-NOVO TRANSCRIPTOME. Avg length, N50, number of annotations, not even the number of transcripts.

Line 278 ERECTA doesn’t seem to be in Fig.3b.

Section 3.2 Line 280 – 286 please remove all discussion points to the discussion, keep this section  for presenting the results (and make sure the transcripts mentioned are in and named the same as in the figure).

Line 294 Typo “were” -> “had”

Line 295 and others “Tv+Ch” presumably these samples are what has previously been named TC in the manuscript (and figures) – unless it really is a combination of the data from the Tv and Ch samples?

Line 297 – 299 “Such a comparison showed the effect of the pathogen on plant growth in the case of T. virens colonization was less strongly affected than a direct injection of C. homoeocarpa” this is a discussion point. I’d dispute the claim as well – certainly singularly infected C.homoeocarpa samples had more DEGs at 19 DPI than coinfection with Trich at 19 DPI.

Line 300 (and 349) mentions Data S2, this was not provided with the ms – is this supposed to be included?

Line 300 - 313 while these all look like interesting results they need to be set into context with how many of the domains given are annotated on the transcriptome – the whole point of doing enrichment analysis. Without this information it’s difficult to judge whether these are random event or significant (you mention the same issue yourselves on line 314). Please give the full methods for enrichment analysis in the methods section

Line 320 – 321 how were the candidate genes for qPCR selected?

Line 322 – 324 These are discussion points, please move.

Metabolome - don’t use up and down regulated use increased/decreased abundance, also DEM – I don’t like the phrase, they’re not expressed.

Line 339 – 340 “Filtering these DEMs by the VIP of PLS-DA and classification after combing HMDB and KEGG databases” I’m not certain what you’ve done here. This needs to be added to the methods section (also add a quick explanation of VIP). The metabolomics methods need improving and give full detail have how (and using which implementation) PLS-DA was carried out.

Line 344 – 346 Metabolites similar to (I assume so anyway)

Line 346 - 348 -> discussion

 Line 356 – 358 “However, the biosynthetic product of L-phenylalanine is up-regulated in the Tv+Ch group, such as phenylpropanoids (i.e., rosarin) contribute to plant responses to abiotic and biotic stresses (Dong and Lin, 2021)” there’s a typo here

Line 381 – 404 this is mostly discussion

Line 392 Typo “Figure 4b” -> “Figure 5b” – please check all of these as well

Discussion

Line 412 – 413 ref needed

Line 419 – 420 “The mechanism of T. virens–As interaction can be viewed in two respects (Figure 6)” ->  ““The mechanism of T. virens-As interaction can be viewed in two respects, plant growth promotion and plant defense response, both discussed below (Figure 6)”

Line 434 Typo “). And” -> “and". Don’t start sentences with and

Line 441 – 443 Please edit this sentence it doesn’t make sense

Line 457 Typo “whose” -> “which”

Line 462 Typo “affected” -> “affects” – don’t use past tense in discussion see line 467 as well (may be others)

Lines 476 – 478 “Besides…” Please edit this sentence – also the figure is incorrect

Line 494 “Besides…” Please edit this sentence it doesn’t make sense

Line 499 “It is consistent” what is consistent?

Line 502 – 503 “under conditions” what conditions?

Line 487 – 511 This is poorly written and needs full editing – as far as I can work it out the points made are worth making.

Line 513 delete “occurs”

Line 512 – 526 This needs rewriting as the meaning of many sentences is not clear.

Figs:

Figure 4 b,d,e are not used in the ms. Please check all sub plots of fig1 and 2 as well. If presented in main figures they need to be talked about in the MS

Figure 1 – Label need improving. Are the treatments in the photo in the same order as the graphs? Relabel giving the full names. The text is really small on the figures – also, I’m not certain you need tan x-axis title as it is so obvious. I don’t understand the y axis units. I can guess the lines indicate significant diff. but what alpha do the stars indicate. Also, I’m not convinced using the connecting lines is any more useful than the traditional method of using letters to indicate sig. groups (actually just spotted the methods section suggests lettered grouping is to be used). I’m not convinced adding these details to graphs is of benefit, but I know plenty this is not a universal opinion (I prefer the was the graphs are presented in fig2…).

Figure2 need to give the names of the treatments in full in the legend – and why does this use lettering rather than the lines as per fig.1. Please be consistent

Fig 3. a. perhaps better as 2 Venn diagrams – and how many treanscipts in total are we talking about here. What is C. the description is terrible at the moment. The legend states FPKM, but it looks like some sort of zero centering has gone on – is this gene wise? And why are there only 3 samples presented when the methods state there were 5 measured?

Fig 4. Text is too small on a,b,d,e  - it’s not readable even after fully zooming in on my monitor. Fig e is problematic – the x axis has 5 x the variability of the y-axis. There’s no good reason to draw the ellipsoids as shown (as the PLS-DA plots are not mentioned in the ms, just get rid of them).  Perhaps split into 2 plots. Abundance -> relative abundance

Fig 5. Text too small again even at full magnification. Top 100 by abundance?

Fig S2 How many annotations of each type in the transcriptome? Also how many transcripts are we talking about. Increased and decreased expression in the same enrichment analysis? Generally, I think it’s better to do analyses for increased and decrease transcripts separately (or at least in addition).

Author Response

Dear Reviewer,

Thank you for your valuable review and suggestions about our manuscript entitled “New Insights into the Mechanism of Trichoderma virens-induced Developmental effects on Agrostis stolonifera Desease Resistance Against Dollar Spot Infection” (ID: 1911343). Those comments are all valuable and important for improving the quality of our work. We have studied the comments carefully and would like to revise them (Note: The line numbers in all answers are modified positions). We have addressed the comments or response it in attachment.

Round 2

Reviewer 3 Report

I would like to thank the authors for considering my recommendations and I am happy with the explanations given where we disagree.

I do still have a few minor points which do need to be addressed.

I'm still not certain how the expression analysis was performed. I am pretty certain RSEM is not used to identify DE transcripts. Possibly something like edgeR or DESeq2 was used? Both of these methods additionally include library size normalization - which is necessary for RSEM abundance estimates. Using FPKM values for library size normalization is not ideal.

I note in one of your comments (26) that transcriptome and metabolome analyses were performed by a 3rd party company, though this doesn't seem to be mentioned in the methodology? Please indicate in the methods which steps were done in house and which were outsourced.

Line 239: "(FC) ≥1.2 or ≤0.83 ... were set as the filtering condition" Surely this can't be right?

The legend on Fig 3 is incorrect. It's not possible to have negative FPKM values.

Fig S2 - I think this would be improved by incorporating the total number of annotations in the transcriptome matching each category into the graph. Either give the total numbers for each category as a figure or as additional bars, or alternatively give the proportion of each annotation in DE transcipts compared to the total (extra bars or just quote the proportion). Without this the figure lacks any context (I do note it is supplemental).

Author Response

Thank you for your valuable review and suggestions again. We have studied the comments carefully and would like to revise them (Note: The line numbers in all answers are modified positions). We have addressed the comments or response it as followed:

  1. I'm still not certain how the expression analysis was performed. I am pretty certain RSEM is not used to identify DE transcripts. Possibly something like edgeR or DESeq2 was used? Both of these methods additionally include library size normalization - which is necessary for RSEM abundance estimates. Using FPKM values for library size normalization is not ideal. I note in one of your comments (26) that transcriptome and metabolome analyses were performed by a 3rd party company, though this doesn't seem to be mentioned in the methodology? Please indicate in the methods which steps were done in house and which were outsourced.

Answer:

We apologize for the lack of details again. Indeed, the differentially expressed genes are detected and calculated with counts by DESeq2, which I have added to the Method 2.4 (Ln 205-209).

Good point. I have added it (RNA-seq: Ln 212-213; Metabolome: Ln 236).

  1. Line 239: "(FC) ≥1.2 or ≤0.83 ... were set as the filtering condition" Surely this can't be right?

Answer: I reconfirmed the report of metabolome, the filtering condition of (FC) ≥1.2 or ≤0.83 is right. The other literatures are also the condition (Morris et al., 2021). As for why, I will continue to explore.

Literature example:

Morris, E. M., Kitts-Morgan, S. E., Spangler, D. M., Ogunade, I. M., McLeod, K. R., & Harmon, D. L. (2021). Alteration of the canine metabolome after a 3-week supplementation of cannabidiol (CBD) containing treats: an exploratory study of healthy animals. Frontiers in Veterinary Science, 8.

  1. The legend on Fig 3 is incorrect. It's not possible to have negative FPKM values.

Answer: I am sorry for the mistake. I have modified it.

  1. Fig S2 - I think this would be improved by incorporating the total number of annotations in the transcriptome matching each category into the graph. Either give the total numbers for each category as a figure or as additional bars, or alternatively give the proportion of each annotation in DE transcipts compared to the total (extra bars or just quote the proportion). Without this the figure lacks any context (I do note it is supplemental).

Answer: Good point. Although, Figure S2 is only a barchart based on kegg class classification of DEGs, it is not the enrichment analysis based on all annotated gene and DEGs (do I understand correctly?). In order to present more clearly, I added a legend to Figure S2 to explain the total number of DEGs for the barchart analysis (the proportion will naturally come out).

I would like to express our sincere gratitude to you again. We hope these responses and revised manuscript could be acceptable for the journal.

Sincerely,

Lu Gan